# Wide-Angle Polarization-Independent Ultra-Broadband Absorber from Visible to Infrared

**DOI:** 10.3390/nano10010027

**Published:** 2019-12-20

**Authors:** Jing Liu, Wei Chen, Jia-Chun Zheng, Yu-Shan Chen, Cheng-Fu Yang

**Affiliations:** 1School of Information Engineering, Jimei University, Xiamen 361021, China; jingliu@jmu.edu.cn (J.L.); jchzheng@jmu.edu.cn (J.-C.Z.); chenys@jmu.edu.cn (Y.-S.C.); 2Navigation Institute, Jimei University, Xiamen 361021, China; 3Department of Chemical and Materials Engineering, National University of Kaohsiung, No. 700, Kaohsiung University Rd., Nan-Tzu District, Kaohsiung 811, Taiwan

**Keywords:** metamaterial, light absorption, localized surface plasmon resonance, nanostructure

## Abstract

We theoretically proposed and numerically analyzed a polarization-independent, wide-angle, and ultra-broadband absorber based on a multi-layer metasurface. The numerical simulation results showed that the average absorption rates were more than 97.2% covering the broad wavelength of 400~6000 nm (from visible light to mid-infrared light) and an absorption peak was 99.99%, whatever the polarization angle was changed from 0° to 90°. Also, as the incidence angle was swept from 0° to 55°, the absorption performance had no apparent change over the wavelength ranges of 400 to 6000 nm. We proved that the proposed metasurface structure was obviously advantageous to achieve impedance matching between the absorber and the free space as compared with conventionally continuous planar-film structures. The broadband and high absorption resulted from the strong localized surface plasmon resonance and superposition of resonant frequencies. As expectable the proposed absorber structure will hold great potential in plasmonic light harvesting, photodetector applications, thermal emitters and infrared cloaking.

## 1. Introduction

Metamaterial is a new type of composite material with sub-wavelength structure, usually composed of metal models and dielectric layers. It can reveal many physical properties not found in traditional materials, such as a negative refractive index [1], electromagnetic cloaking [2,3], perfect absorption [4], and ultra-sensitive sensing [5]. Since the first perfect metamaterial absorber was reported by Landy et al., the direction of metamaterial absorbers had been studied by many scholars. To date, the absorption ranges of the investigated absorber have covered the spectra from visible light to microwave frequency [6,7,8,9]. Extraordinary, broadband absorbers in the range of visible and middle infrared light have been widely investigated for many promising applications, such as solar cells [10], thermal imaging [11], and thermal detection [12], and so on. However, there are many common shortcomings in most of the reported absorbers such as low absorption, narrow absorption broadband, and polarization-dependent, which seriously restricts their practical applications. In order to get perfect broadband absorption, different designed methods are proposed recently, including using patterned metallic arrays [13], multilayer structures [14], and planar films without structure patterning [15]. Among these schemes, some of them usually suffer from a complicated fabrication process or a limited absorption bandwidth. Besides, noble metals (e.g., gold and silver) and certain complex patterned structures are widely used in many absorber designs due to their excellent surface plasmon polarization, which results in expensive cost in the practical productions [16]. Thus, it is still a challenge to design an electromagnetic metamaterial absorber with simultaneous ultra-broadband absorption, fabrication simplicity, and low cost.

The simplest Electromagnetic (EM) absorber could be designed by planar metal/dielectric stacks that require very little nanofabrication, which has constructed a Fabry–Pérot cavity or with the planar surface plasmon. Considering the feasibilities of design and fabrication, the simplest and effective method to broaden the absorption bandwidth is incorporating multiple resonators within a unit cell, even a single resonator is inherent narrowband. An alternative structure to broaden the absorption bandwidth is to design a metasurface, which stacks the multi-layer dielectric and metal with different areas and different morphologies on a metallic substrate. In the past, Cong et al. used one-dimensional multilayered Au/SiO_2_ films via hybridization of propagating surface plasmon to design a broadband visible-light absorber, and the average absorption over the visible band of 400–750 nm could reach only nearly 90% [17]. Zhong et al. proposed a configuration of polarization-selective metamaterial perfect absorber with ultra-wide absorption bandwidth [18]. Wu et al. first proposed that Fe had the beneficial impedance matching effect between the metamaterial structure and the free space, which could be used to construct an ultra-broadband near-perfect solar absorber in the visible and near-infrared region (from 400 to 2000 nm) [19]. Also, there are many researches had investigated different methods or structures to design the ultra-broadband absorber in the visible and near-infrared region [20,21].

In this paper, a broadband absorber is technologically designed and analyzed from visible light to mid-infrared light (400~6000 nm). We proposed a novel broadband absorber consisted of a top Ti circular disk layer, one single SiO dielectric layer, two Fe-SiO layers, and bottom Fe mirror layer. The designed absorber was constructed based on localized surface plasmon resonance (LSPR) by using a multi-layer nanostructured thin layer. However, we would show that the designed absorbers offered better absorption effect (wider bandwidth) than all previously reported broadband absorbers operating in a spectrum from visible light to middle-infrared [10,11,12,13,14,15,16,17,18,19,20,21]. The proposed ultra-broadband absorber could achieve a nearly perfect absorption covering a broad range, the absorption peak was 99.99%, the average absorption rate was 97.2% from visible light to middle-infrared light (i.e., from 400 to 6000 nm), and an absorption bandwidth over 5400 nm (>90%).We would first prove that the structure is advantageous for impedance matching with free space as compared with the reported conventional planar-film structures, which extremely impacts on generating an ultra-broadband perfect absorption. Also, electric field distribution was analyzed for understanding the absorption mechanism. Analysis manifested that the superposition of the resonant frequencies of the top metallic disk and the square metallic film result in near-perfect broadband absorption. When the incident angle was lower than 55°, because the designed structure was symmetry, then the absorber maintained high performance and exhibited a polarization-independent characteristic. In order to further study its absorption performance, the effects of different top disk dielectric materials on the absorption characteristics were also investigated for comparison. Additionally, we also analyzed the number of the metallic-dielectric layers on the absorption characteristics. These analysis results can prove that the proposed structure can realize for broadening absorption bandwidth and enhancing absorption rate. Our proposed absorber will provide a more flexible design and a broader absorption range, which greatly reduces manufacturing costs and structural thickness. As expectable the proposed absorber structure will hold great potential in plasmonic light harvesting, photodetector applications, and infrared cloaking.

## 2. Geometry, Parameters and Methods

The illustration of the proposed broadband polarization-independent absorber is manifested in Figure 1. Figure 1a illustrates the side view of the unit cell of our proposed absorbers which was composed of top Ti top circular disk layer (symbolized l in Figure 1), SiO layer, two iron (Fe)-SiO layers, and bottom Fe mirror layer. It was important to note that the metallic layers weren’t planar and continuous, they were square layers with different widths. In order to clearly observe the structure, the unit cell of propose absorber was divided into three parts, as Figure 1b shows, the square Fe layers were stacked on the SiO layers and they didn’t cover the whole region of SiO layers (their square widths were different). EM absorbers designed using LSPR are relatively simple and the phenomena of absorption can be easily analyzed using the basal principles of interference optics, plasmonics, and metamaterials. For that, the proposed absorber was calculated and optimized numerically based on the Finite-Difference Time-Domain (FDTD) method [22]. In simulation, periodic boundary conditions were applied in the *x*-axis and *y*-axis, and perfectly matched layer boundary condition was used along *z*-axis. A planar broadband linearly polarized wave was with TM polarization propagates along the negative *z*-axis and with the electric and magnetic fields polarized along the x and y directions, respectively. As the incident light was illuminated on the designed structure, the LSPR could be excited in the thin SiO layer between Ti and Fe layers, which could produce a stronger near field in the confined nanospace. The diameter and thickness of Ti and the thicknesses of SiO layer and two Fe-SiO layers were properly optimized to better confine the near field. The period P is set at 300 nm, for that the designed absorber is a sub-wavelength structure for the incident wavelength. The diameter (R) of top disk is 140 mm. The top square layer has a width (w1) of 220 nm and below square layer has a width (w2) of 280 nm. Thicknesses of the absorber from top to bottom are 155 nm (l), 170 nm (h1), 10 nm (h2), 170 nm (h3), and 300 nm (h4), respectively. The spectral absorption rate of the absorber can be obtained by Absorption (A) = 1 − reflection (R) − transmission (T). As the thickness of bottom metal substrate is thicker than the skin depth of the optical range of visible light to infrared region, the transmission of the proposed structure is zero. Therefore, the absorption could be represented as A = 1 − R. SiO could be recognized as the lossless material with a refractive index (n) of 1.53 and the optical parameters of the metal Ti and Fe were taken from CRC *Handbook of Chemistry & Physics* [23]. With respect to an experimental demonstration, the metasurfaces can be manufactured by using three-dimensional electron beam lithography with water ice [24].

## 3. Results and Discussions

The absorption spectrum of the proposed metasurface structure is depicted in Figure 2a, even the absorption rate in the range of 563~647 nm was slightly lower than 90% (the lowest absorption was 88.5%), the designed absorber revealed a nearly perfect absorption (over 95%) in the range of 1085~5690 nm. The average absorption is calculated as $A=\int_{\lambda 1}^{\lambda 2}A\left(\lambda \right)d\lambda /\left(\lambda_{1}-\lambda_{2}\right)$, where λ_1_ and λ_2_ are 6000 and 400 nm, respectively. According to our calculation, the average absorption rate of proposed absorber reached 97.2% in the wavelength range of from 400 to 6000 nm, which had obviously exceeded most of the previously reported absorbers, both in both absorption efficiency and perfect absorption bandwidth [10,11,12,13,14,15,16,17,18,19,20,21]. To illustrate the significance and novelty of the proposed absorber, main characteristics of state-of-the-art absorbers from visible to infrared lights are listed in Table 1 for comparisons. As we all know, high absorption rate in wide bandwidth is one of the most important indexes of the designed absorbers. Also, the designed absorbers being independent of angular stability and polarization are necessary in practical applications. The materials and the numbers of layers are directly related to their production costs and manufacturing processes, so they also need to be considered. One can see that from Table 1 that our proposed absorber could offer wider absorption bandwidth from visible to infrared light. The most important thing is that our proposed absorber is low cost (with no noble metal) and has ultra-thin thickness. To additionally verify the advantages of the investigated metasurface, we compared the absorption spectrum of the metasurface structure with that of planar structure (planar structure), where the all other parameters were identical to those of the proposed one. The absorption spectra of two cases are compared in Figure 2b, where the red dotted showed the absorption of the planar structure. We had found that within the analyzed range, the absorption bandwidth of the metasurface structure is much broader than that of the continuous planar film.

To investigate the design model in Figure 3, we first calculated the structure A shown in Figure 3a. The absorption spectra versus different values of w2 and h2 are shown in Figure 3b,c. From Figure 3b, one can see that as w2 increased, the absorption range of the designed structure was significantly widened. However, when w2 had a value of 280 nm, the absorption efficiency of the structure significantly decreased. Thus, as w2 was equal to 280 nm, the designed structure could construct a good absorber. Figure 3c shows the absorption spectra as a function of h2. When the h2 was changed from 5 to 30 nm, the absorption range significantly decreased, and the absorption efficiency significantly increased for wavelength being changed from 2500 to 4000 nm. Moreover, when h2 = 10 nm, the absorption efficiency and absorption range revealed a high level at the same time. Thus, h2 was determined to be 10 nm. Then, in Figure 4 we calculated the absorption spectra structure B shown in Figure 4a. When the w1 was changed from 180 to 280 nm, the absorption range of the structure significantly increased, however, the absorption efficiency decreased as the wavelength was changed from 1000 to 4000 nm. When w1 = 240 nm, the designed structure reveals the high absorption efficiency and the broad absorption range. For that w1 is determined to be 10 nm. Finally, structure C shown in Figure 5a was also analyzed by changing the designed parameters and l and h1 were set at 155 nm and 170 nm, respectively. In all, we can determine the optimal parameters by observing the absorption spectra.

To further understand the behavior of the high absorption rate in the broadband wavelength range, we used the impedance matching method to calculate and analyze the proposed absorber in detail. As we know, impedance matching result plays an extremely important role for generating a perfect absorber, for that effective medium theory is utilized to analyze the impedance matching condition. The relation between impedance *Z* and *S* parameters can be expressed as [25,26]:(1)$$S_{11}=S_{22}=\frac{i}{2}\left(\frac{1}{Z}-Z\right)\sin\left(nkd\right),$$
(2)$$S_{21}=S_{12}=\frac{1}{cos\left(nkd\right)-\frac{i}{2}\left(Z+\frac{1}{2}\right)sin\left(nkd\right)},$$
where *S*_11_, *S*_22_, *S*_21_, and *S*_12_ are *S* parameters and *n*, *k*, and *d* are the effective refractive index, the wave vector, and the thickness of designed structure, respectively. Therefore, the impedance *Z* is yielded by:(3)$$Z=\pm \sqrt{\frac{{\left(1+S_{11}\right)}^{2}-S_{21}^{2}}{{\left(1-S_{11}\right)}^{2}-S_{21}^{2}}},$$

As we know that if we would design a broadband perfect absorber, the impedance over an ultra-broad frequency band in the visible and middle infrared region should match the free-space impedance (*Z* = *Z*_0_). According to Figure 6, clearly, the impedance of real part is close to 1 and imaginary part is close to 0 for a light range from 1400 to 4900 nm, which agrees well with perfect absorption effect with the broad bandwidth, as the result in Figure 1a shows.

To obtain the further insights of the physical mechanisms for the nearly perfect broadband absorption, we also calculated the electric field intensity distribution (|E|^2^) at the wavelengths of λ = 500, 1500, 2500, 3500, 4500, and 5500 nm, and the results are shown in Figure 7. From these figures, we could find that there existed the effect of localized electrical field enhancement around the two edges of the top disk and two metallic layers, which were caused by the different resonant wavelengths. The absorption in shorter wavelengths (@ 500 nm) is mainly caused by the oscillations located in upper Ti disk. With increasing the wavelength of incident light, the electromagnetic oscillations located in the upper Ti disk become weaker, but the oscillations located in lower Fe layer are enhanced with increasing operating wavelength. In addition, the wider Fe layer could support the electromagnetic oscillations at longer wavelengths. These results prove that the lower metallic layer could support the electromagnetic oscillations at longer wavelengths and the enhancement behavior of strong LSPR and super positions of resonant frequencies lead to the strong absorption in such a broadband region.

## 4. Discussions

It should be noted that in the above discussion, incident light is incident vertically and polarized along the *x*-axis direction. But in practical applications, the incident angle and the polarization angle need to be considered. Therefore, in Figure 8 we further investigated the absorption rate of the proposed absorber by changing the incident angle θ and polarization direction φ. The thing needed illustration is that we used the Bloch boundary conditions for simulating the oblique incident light in the *x*-axis and *y*-axis directions and the broadband fixed-angle source technique was used to ensure that all the wavelengths would have the exactly incident angle. From Figure 8a, one can see that there was almost no apparent change for the absorption performance when the polarization directions (φ) were changed from 0° to 90°. Independent polarization is due to the symmetry of the proposed structure. The relationship between absorption spectrum and incident angles θ was shown in Figure 8b. When the angle of incidence was swept from 0° to 55°, the absorption spectra had no apparent change over the wavelength ranges of 400~6000 nm, show that absorber can be applied even the incident light is in a large angle. These results suggest that the proposed absorber is a broadband, polarization-independent, and wide-angle nearly perfect absorber from visible light to mid-infrared light.

For further proving the effect of the used top metal on the absorption performances, we also analyzed the effects of different metal on the absorption spectra and the absorption spectra are plotted in Figure 9a for different metals, including Ti, chromium (Cr), and gold (Au). We had apparently found that the top metal had large effect on the absorption spectra in the region of visible to near-infrared light (i.e., from 400 to 1500 nm). When Au disk was used as top metal, absorption rate from visible to near-infrared was very low. Also, using Cr top metal the absorption rate is smaller than that of Ti metal in the discussed region of visible to near-infrared light. These results prove that the absorption performance in the visible region (400 to 1500 nm) is closely related to the top metal, which matches the electric field distributions (@500 nm, @1500 nm) in Figure 7. Ti is the optimal metal because it can reach the highest absorption rate in the range of 400 to 1500 nm. Since Ti is a highly lossy metal, the Q-factor of the LSPR is rather low, which is good for high absorption [20]. For further proving that the SiO could construct a better absorber, we also analyzed the influence of dielectric-layer materials on the absorption characteristics. Figure 9b shows the absorption spectra versus different dielectric layers. When dielectric layers were SiO_2_ (n = 1.45), SiO (n = 1.53), and Al_2_O_3_ (n = 1.73) [27], the absorption of absorber would decline to 90% at 5680 nm, 5889 nm, and 6080 nm, respectively. As compared with those of SiO_2_ and SiO films, although the average absorption rate and the absorption peak of Al_2_O_3_ film from 400 to 5400 nm had the lowest value, but it had the broadest absorption bandwidth. The function of the dielectric layers is that they can separate the metal layers to form the superposition of resonant frequencies. Moreover, the dielectric layers with different refractive index (n) can influence the optical properties which can flexibly adjust the absorption phenomenon. When incident light is from the air into the dielectric layers, it will reflect from or transmit them, which is strongly dependent on the n values of dielectric layers. As n value changes, the reflection and transmission rates of the absorber will be different as the wavelengths of incident light is changed. In addition, n values of the dielectric layers also affect the interaction effect existed between electrons in metal and incident light [28]. These results also prove that the broad bandwidth is closely related to the used dielectric material and SiO is the optimal dielectric material. Even using SiO as the dielectric material cannot have the maximum bandwidth in the middle infrared region, it can have the highest absorption rate in the ranges of 400 to 6000 nm. In this study, SiO is recognized as the lossless material with a refractive index of 1.53, and it is the optimal choice.

Modern nanotechnology opens up possibilities to combine the nanostructures of various materials with very different characteristics in one superstructure, the surface reconstruction caused by the structural defects cannot be ignored. Zhang et al. theoretically studied the novel optical properties of hybrid molecules composed of semiconductor and metal nanoparticles [29]. Later, hybrid excitons along with new opportunities due to impurities or structural defects have been extensively studied by many scholars [30,31,32,33]. In this paper, the possible defect due to the deposition on a substrate should be considered for experiment. To simulate structural defects, we set that the iron layers is uneven in Figure 10. So the gaps in the iron layers are filled with SiO. Absorption spectrum of the proposed normal model compared to that of the defective model is plotted in Figure 10c, and the results show that there is almost no difference for the two absorption spectra. We also calculated in Figure 11 and Figure 12, which show the distributions of electric fields of different models at the different wavelengths. One can easily see that although the defect affects the electric field, but the effect is much weaker than LSPR. Thus, minor structural defects do not significantly affect the overall optical performance of the proposed structure, which has a high tolerance for structural defects.

## 5. Conclusions

In this paper, a novel ultra-broadband absorber based on metasurface was technologically designed and analyzed using the FDTD method in the ranges of visible light to middle infrared light (about 6000 nm). The bandwidth of absorption rate over 90% could reach 5500 nm, the absorption peak was 99.99%, and the average absorption rate was 97.2% from ranges of visible light to middle-infrared light (400~6000 nm). We had proven that the broadband effect and the high absorption mainly result from the strong localized surface plasmon resonance (LSPR) and superposition of resonant frequencies. The proposed absorber was polarization-independent and wide-angle (below 55°), which had the characteristics of nearly perfect absorption. The effects of used metals and dielectric materials on absorption performance were also analyzed in detail. We have found that Ti is the optimal metal because it can reach the highest absorption rate in the range of 400~1500 nm and SiO is the optimal dielectric material because it can have the highest absorption rate in the ranges of 400 to 6000 nm, which offer better absorption than much previously reported broadband absorber operating in a spectrum from visible light to middle-infrared. However, our proposed absorber is low cost and has ultra-thin thickness and stable, it can easily find the applications in spectral imaging, solar energy harvesting, and thermo-photovoltaic energy conversion.

## Figures and Tables

**Figure 1 nanomaterials-10-00027-f001:**
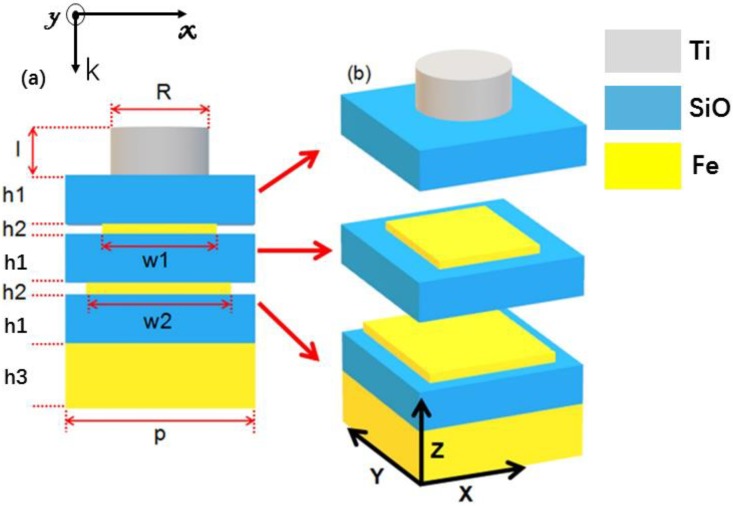
Schematic of the absorber design. (**a**) Side view of a unit cell of the absorber, the optimum parameters were p = 300 nm, l = 155 nm, R = 140 nm, h1 = 170 nm, h2 = 10 nm, h3 = 300 nm, w1 = 240 nm, and w2=280 nm. (**b**) Demonstration of the structure with a unit cell, in which each layer is separated from each other for clear observation.

**Figure 2 nanomaterials-10-00027-f002:**
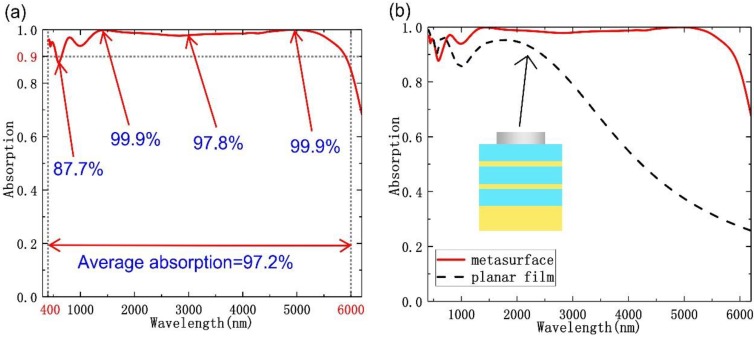
(**a**) Absorption spectrum of proposed absorber. (**b**) Absorption spectrum of the proposed metasurface absorber (red solid line) compared to that of the planar structure (black dash-dot line).

**Figure 3 nanomaterials-10-00027-f003:**
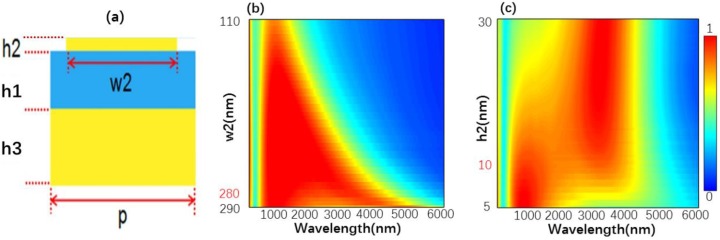
(**a**) Side view of a unit cell of the structure A, p = 300 nm, h1 = 170 nm, h3 = 300 nm. Absorption spectra versus different values of (**b**) w2 and (**c**) h2.

**Figure 4 nanomaterials-10-00027-f004:**
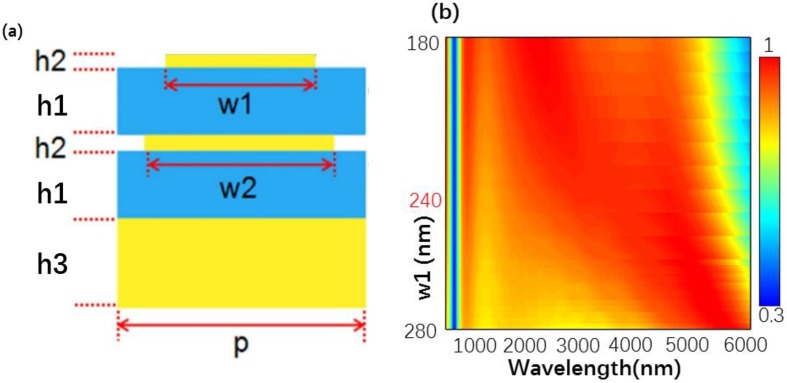
(**a**) Side view of a unit cell of the structure B, p = 300 nm, h1 = 170 nm, h2 = 10 nm, h3 = 300 nm, and w2 = 280 nm (**b**) Absorption spectra versus different value of w1.

**Figure 5 nanomaterials-10-00027-f005:**
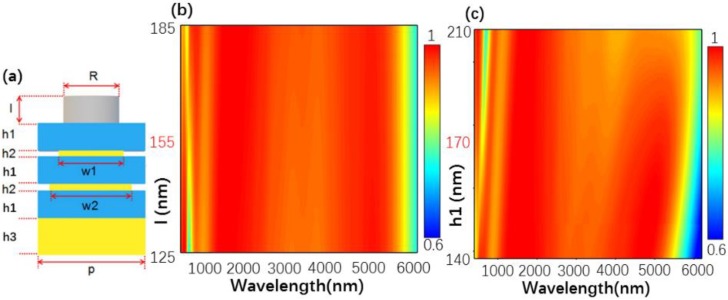
(**a**) Side view of a unit cell of the structure C, p = 300 nm, R = 140 nm, h2 = 10 nm, h3 = 300 nm, w1 = 240 nm, w2 = 280 nm. Absorption spectra versus different values of (**b**) l and (**c**) h1.

**Figure 6 nanomaterials-10-00027-f006:**
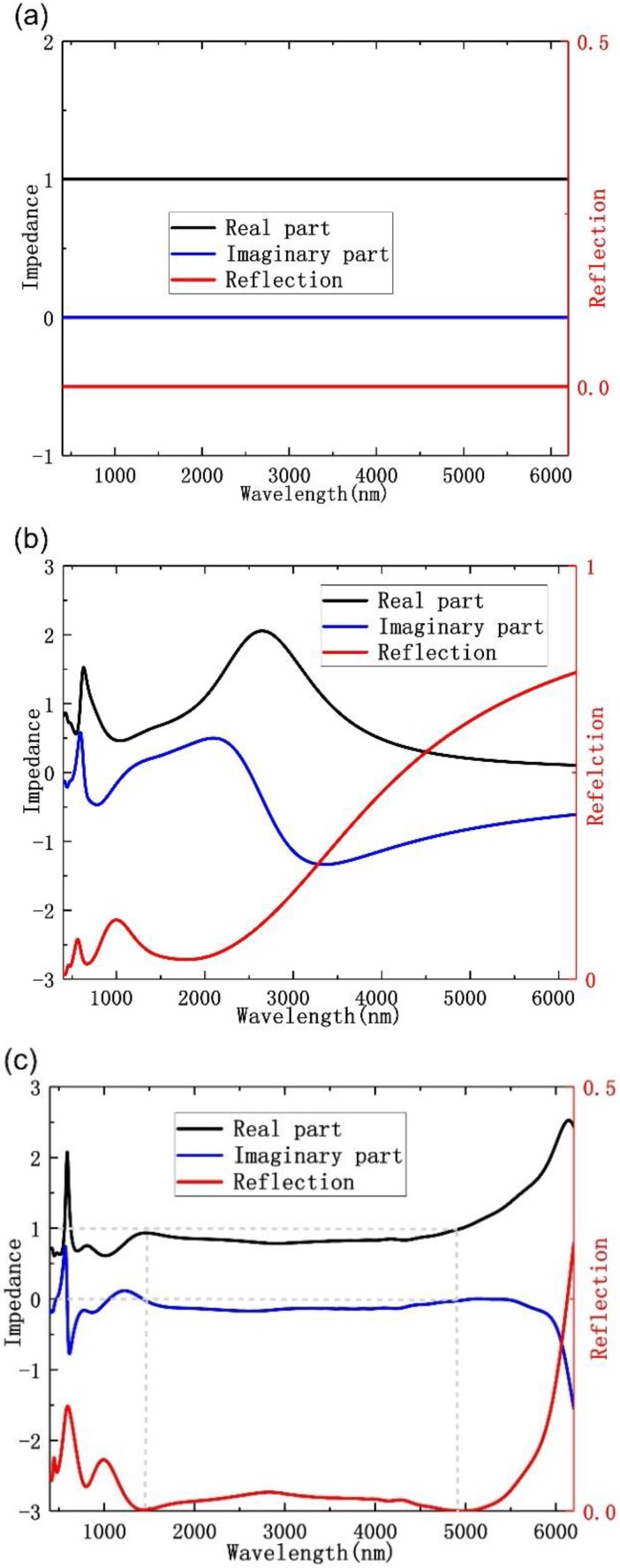
Impedance curve and reflection of the (**a**) air, (**b**) planar film structure, and (**c**) metasurface structure.

**Figure 7 nanomaterials-10-00027-f007:**
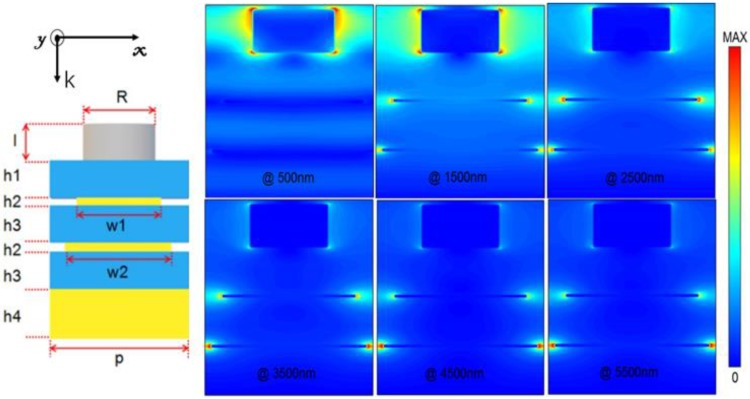
Electric field distributions at different wavelengths.

**Figure 8 nanomaterials-10-00027-f008:**
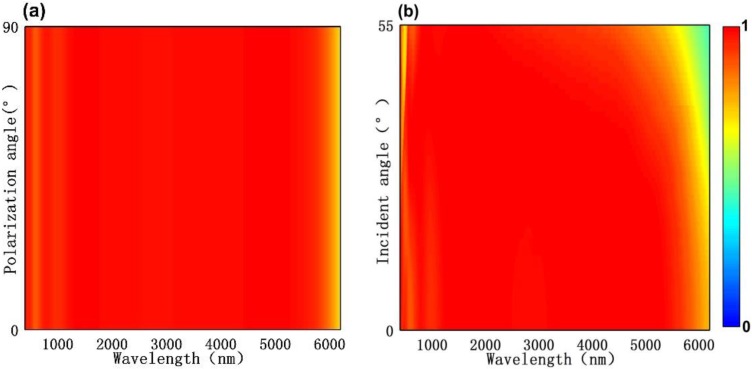
Absorption spectra versus different (**a**) polarization angle φ and (**b**) incidence angle θ.

**Figure 9 nanomaterials-10-00027-f009:**
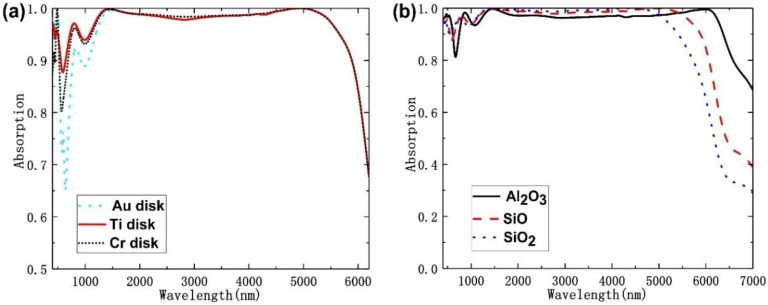
Absorbance spectra versus (**a**) different top metals and (**b**) different dielectric layers.

**Figure 10 nanomaterials-10-00027-f010:**
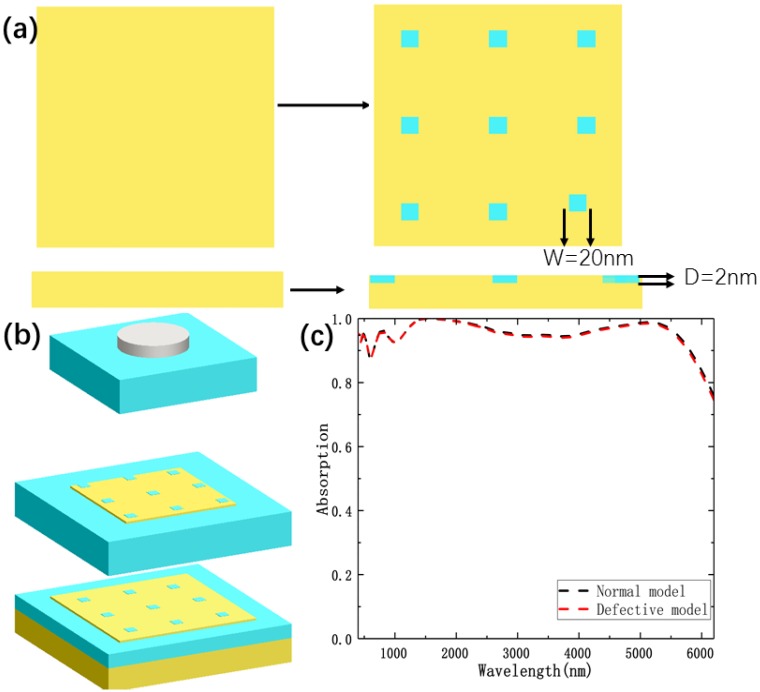
(**a**) A schematic diagram of defects in the iron layers. In two Fe layers, nine small holes (W = 20 nm and D = 2 nm) were filled with SiO. (**b**) Schematic of the defective model design. (**c**) Absorption spectrum of the proposed normal model compared to that of the defective model.

**Figure 11 nanomaterials-10-00027-f011:**
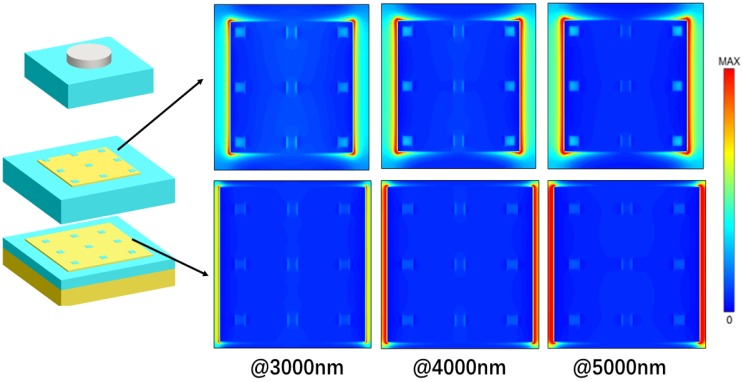
Electric field distributions of the defective model at different wavelengths.

**Figure 12 nanomaterials-10-00027-f012:**
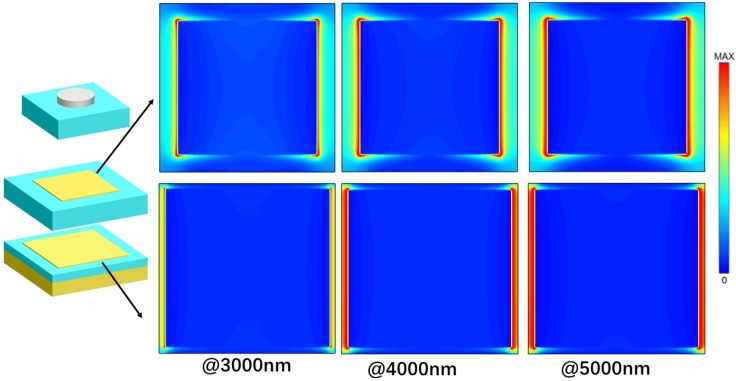
Electric field distributions of the normal model at different wavelengths.

**Table 1 nanomaterials-10-00027-t001:** Comparisons between plasmonic absorbers from visible to infrared.

Reference	Bandwidth *	Angular Stability	Polarization Independent	Materials	Number of Layers
[12]	~2800 nm	up to 60°	NO	Au, Ge	41
[14]	~1700 nm	up to 70°	NO	SiC, Ag	41
[18]	~4000 nm	up to 45°	NO	Al, SiO_2_, Ti	33
[21]	~3000 nm	up to 50°	NO	Ti, Al_2_O_3_	6
this work	~5500 nm	up to 60°	YES	Ti, SiO, Fe	7

* Bandwidth of absorption above 90%.

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
