# Peer review of "Wide-Angle Polarization-Independent Ultra-Broadband Absorber from Visible to Infrared"

_nanomaterials, 2019, doi:10.3390/nano10010027_

Round 1

Reviewer 1 Report

Manuscript ID: nanomaterials-667660

Title: Wide-Angle Polarization-Independent Ultra-Broadband Absorber from
Visible to Infrared
Authors: Jing Liu, Wei Chen, Jiachun Zheng, Yushan Chen, Chengfu Yang

The authors study nanosized heterostructures consisting of metallic
and dielectric layers in order to design structures with maximal
absorption strength in a frequency range from infrared to visible light.
According structures (termed metamaterials) are currently heavily studied
for various applications. Broadband absorbers in the infrared and vissible
range have promising applications in thermal sensing and imaging and might
improve solar cells design.

The authors propose a structures formed by seven different layer. They
determine the frequency dependent absorption coefficient using the
finite-difference time-domain method to solve the according electrodynamic
equations. They observe an absoprtion above 90% for most of the investigated
frequency window with minimal dependence on light polarization or incident
angle.

The paper is clearly written, the results are interesting and well described.
I hesitate to recommend publication at the moment for the following reasons:

1.) The authors discuss a single design of seven layers with given width
and thickness. I would expect a thorough investigation by varying the
thicknesses of the layers and testing different basic designs but the
manuscript does not comment on an such investigation. So, the question
remains: why this design ?

2.) The introduction mentions several other works trying to optimize
broadband absorbers but the discussion section does not give an in depth
comparisson to show where and by how far this new design is better.

Minor point:

3.) All materials have defects, for example, structural defects due to
surface reconstruction or molecular defects due to impurities. By
designing a broadband absorber using surface plasmon resonance such
defects might strongly influence the overall optical performance of the
device by the strong defect plasmon interaction (see, f.e.
Phys. Rev. Lett. 2006, 97, 146804; J Phys Chem Lett. 2016 Jun 2;7(11)).
It would be nice to comment on this.

Author Response

Response to Reviewer 1 Comments

Point 1: The authors discuss a single design of seven layers with given width and thickness. I would expect a thorough investigation by varying the thicknesses of the layers and testing different basic designs but the manuscript does not comment on a such investigation. So, the question remains: why this design?

Response 1: Thanks for your comment, please lines 153-178

     To investigate the design model in Figure 3, we first calculated the structure A shown in Figure 3(a). The absorption spectra versus different values of w2 and h2 are shown in Figures 3(b) and 3(c). From Figure 3(b), one can see that as w2 increased, the absorption range of the designed structure was significantly widened. However, when w2 had a value of 280 nm, the absorption efficiency of the structure significantly decreased. Thus, as w2 was equal to 280 nm, the designed structure could construct a good absorber. Figure 3(c) shows the absorption spectra as a function of h2. When the h2 was changed from 5 to 30 nm, the absorption range significantly decreased, and the absorption efficiency significantly increased for wavelength being changed from 2500 to 4000 nm. Moreover, when h2=10 nm, the absorption efficiency and absorption range revealed a high level at the same time. Thus, h2 was determined to be 10 nm. Then, in Figure 4 we calculated the absorption spectra structure B shown in Figure 4(a). When the w1 was changed from 180 to 280 nm, the absorption range of the structure significantly increased, however, the absorption efficiency decreased as the wavelength was changed from 1000 to 4000 nm. When w1=240 nm, the designed structure reveal the high absorption efficiency and the broad absorption range. For that w1 is determined to be 10 nm. Finally, structure C shown in Figure 5(a) was also analyzed by changing the designed parameters, where l and h1 were set at 155 nm and 170 nm, respectively. In all, we can determine the optimal parameters by observing the absorption spectra.

Point 2: The introduction mentions several other works trying to optimize broadband absorbers but the discussion section does not give an in depth comparison to show where and by how far this new design is better.

Response 2: Thanks for your suggestion, please see lines 132-140 and Table 1.

To illustrate the significance and novelty of the proposed absorber, main characteristics of state-of-the-art absorbers from visible to infrared lights are listed in Table 1 for comparisons. As we all know, high absorption ratio in wide bandwidth is one of the most important indexes of the designed absorbers. Also, the designed absorbers being independent of angular stability and polarization are needed consideration in practical applications. The materials and the number of layers are directly related to their production costs and manufacturing processes, so they need to be considered. One can see that from Table 1 that our proposed absorber could offer wider absorption bandwidth from visible to infrared light. The most important thing is that our proposed absorber is low cost (no noble metal) and has ultra-thin thickness.

Point 3: All materials have defects, for example, structural defects due to surface reconstruction or molecular defects due to impurities. By designing a broadband absorber using surface plasmon resonance such defects might strongly influence the overall optical performance of the device by the strong defect plasmon interaction (see, f.e. Phys. Rev. Lett. 2006, 97, 146804; J Phys Chem Lett. 2016 Jun 2;7(11)). It would be nice to comment on this.

Response 3: Thanks for your comment, please lines 268-291.

Modern nanotechnology opens up possibilities to combine the nanostructures of various materials with very different characteristics in one superstructure, the surface reconstruction caused by the structural defects cannot be ignored. Zhang et al. theoretically studied the novel optical properties of hybrid molecules composed of semiconductor and metal nanoparticles [29]. Later, hybrid excitons along with new opportunities due to impurities or structural defects have been extensively studied by many scholars [30-33]. In this paper, the possible defect due to the deposition on a substrate should be considered for experiment. To simulate structural defects, we set that the iron layers is uneven in Figure 10. So the gaps in the iron layers are filled with SiO. Absorption spectrum of the proposed normal model compared to that of the defective model is plotted in Figure 10(c), and the results show that there is almost no difference for the two absorption spectra. We also calculated in Figures 11 and 12, which show the distributions of electric fields of different models at the different wavelengths. One can easily see that although the defect affects the electric field, but the effect is much weaker than LSPR. Thus, minor structural defects do not significantly affect the overall optical performance of the proposed structure, which has a high tolerance for structural defects.

Reviewer 2 Report

The article discusses an interesting metasurface design, which allows one to achieve a high absorption coefficient in a very wide spectral range and, as the authors believe, in a wide range of incidence angles, regardless of the state of polarization of light. From a practical point of view, such a structure would be very useful for optimizing solar energy absorbers, for thermophysical measurements, etc. However, some doubts are nevertheless caused by the presented results regarding the absorption of the proposed surface with variation in the angle of incidence and with variation in polarization for incident light. It was stated at the beginning of the article (line 97) that “In all simulation, periodic boundary conditions were applied in the x-axis and y-axis”, when discussing the calculation method. It is well known that errors can appear, when periodic boundary conditions are used for systems where the structure is periodic, but the EM fields are not.  In particular, this is the case when a periodic structure is illuminated by a plane wave propagating at an angle.  The fields will not be quite periodic in this case, as there will be a phase difference between each period of the device.  It is not clear from the text whether the authors took this phase difference into account and whether they introduced the corresponding Bloch correction to the boundary conditions?

In addition, the phrase (lines 206–208): “The phenomenon can be explained by changing the interaction between electrons in metal and incident light, which is strongly dependent on its dielectric-surrounding medium [27],” seems to be suitable for explanations of any phenomena related to plasmonics and, therefore, in the particular case under consideration explains little. Naturally, the authors have the results of numerical simulations in hand, but if they want to offer some physical explanation for the fact that SiO is the optimal choice, this explanation should be much deeper.

Finally, Fig. 1 would be much better perceived by the reader if the materials from which the structure is made would be directly indicated in this figure. Moreover, instead of designations of sizes p, l, R, h1 ... the numerical values of these sizes (since they are constant) would be directly indicated, rather than in the caption to the picture.

Author Response

Response to Reviewer 2 Comments

Point 1: It was stated at the beginning of the article (line 97) that “In all simulation, periodic boundary conditions were applied in the x-axis and y-axis”, when discussing the calculation method. It is well known that errors can appear, when periodic boundary conditions are used for systems where the structure is periodic, but the EM fields are not. In particular, this is the case when a periodic structure is illuminated by a plane wave propagating at an angle. The fields will not be quite periodic in this case, as there will be a phase difference between each period of the device. It is not clear from the text whether the authors took this phase difference into account and whether they introduced the corresponding Bloch correction to the boundary conditions?

Response 1: Thanks for your comment, please lines 221-224.

The thing needed illustration is that for simulating the oblique incident light we used the Bloch boundary conditions in the x-axis and y-axis directions and the broadband fixed-angle source technique, which can ensure that all the wavelengths will have the exactly incident angle.

Point 2: In addition, the phrase (lines 206–208): “The phenomenon can be explained by changing the interaction between electrons in metal and incident light, which is strongly dependent on its dielectric-surrounding medium [27],” seems to be suitable for explanations of any phenomena related to plasmonics and, therefore, in the particular case under consideration explains little. Naturally, the authors have the results of numerical simulations in hand, but if they want to offer some physical explanation for the fact that SiO is the optimal choice, this explanation should be much deeper.

Response 2: Thanks for your comment, please lines 253-264.

The function of the dielectric layers is that they can separate the metal layers to form the superposition of resonant frequencies. Moreover, the dielectric layers with different refractive index (n) can influence the optical properties which can flexibly adjust the absorption phenomenon. When incident light is from the air into the dielectric layers, it will reflect from or transmit them, which is strongly dependent on the n values of dielectric layers. As n value changes, the reflection and transmission ratios of the absorber will be different as the wavelengths of incident light is changed. In addition, n values of the dielectric layers also affect the interaction effect existed between electrons in metal and incident light [28]. These results also prove that the broad bandwidth is closely related to the used dielectric material and SiO is the optimal dielectric material. Even using SiO as the dielectric material cannot have the maximum bandwidth in the middle infrared region, it can have the highest absorption rate in the ranges of 400 to 6000 nm. In this study, SiO is recognized as the lossless material with a refractive index (n) of 1.53, and it is the optimal choice.

Point 3: Finally, Figure 1 would be much better perceived by the reader if the materials from which the structure is made would be directly indicated in this figure. Moreover, instead of designations of sizes p, l, R, h1 ... the numerical values of these sizes (since they are constant) would be directly indicated, rather than in the caption to the picture.

Response 3: Thanks for your comment, we added the materials’ indexes in Figure 1, please see lines 118. Because the structure’s parameters (e.g. l, h1, h2) are not constant, so we added the “the optimum parameters were” in the Figure caption, please wee lines 112-122.

Reviewer 3 Report

The paper report a numerical method for the realization of polarization-independent, wide-angle, and ultra-broadband absorber based on a multi-layer metasurface. The broadband absorber is designed and analyzed by numerical simulations based on Finite-Difference Time-Domain (FDTD) method.

The argument is quite interesting and is worth of investigation. The presentation is good with proper argumentation, despite some language errors in the discussion.

Just few observations:

The choice of Fe in the h3 and h4 layer is not argued properly. In the description of figure 4 it is not clear if the light is incident from the upper or lower part of the metasurfaces and what is the mentioned x-axis. Moreover, have the authors thought how the technological realization of the metasurfaces can be carried out? The multilayer metasurface have to be realized necessarily on a substrate (glass or silicon or something else). Have they thought to the possible drawback due to the deposition on a substrate. Finally, English should be improved in lines: 79-80,104-105, 120-121,

Minor revision of the paper is required before accepting the paper for publication.

Author Response

Response to Reviewer 3 Comments

Point 1: The choice of Fe in the h3 and h4 layer is not argued properly. In the description of figure 4 it is not clear if the light is incident from the upper or lower part of the metasurfaces and what is the mentioned x-axis.

Response 1: Thanks for your suggestion, please lines 56-59 for the reason to use Fe in the h3 and h4 layers. We optimized the expression and modified the Figure 4 to make it more understandable.

Point 2: Moreover, have the authors thought how the technological realization of the metasurfaces can be carried out? The multilayer metasurface have to be realized necessarily on a substrate (glass or silicon or something else). Have they thought to the possible drawback due to the deposition on a substrate.

Response 2: Thanks for your comment, the technological realization of the metasurfaces can be carried out using “three-dimensional electron beam lithography with water ice”, please lines 115-117.  The possible structural defects for the multilayer metasurfaceare discussed in lines 268-281, please check them.

Point 3: Finally, English should be improved in lines: 79-80, 104-105, 120-121,

Response 3: Thanks for your comment, we have improved the English. Please lines 81-84, 106-108, and 125-128.

Round 2

Reviewer 2 Report

The new version of the article looks much better than the previous one. It can be published in present form.